# Percutaneous Irreversible Electroporation for Treatment of Small Hepatocellular Carcinoma Invisible on Unenhanced CT: A Novel Combined Strategy with Prior Transarterial Tumor Marking

**DOI:** 10.3390/cancers13092021

**Published:** 2021-04-22

**Authors:** Feng Pan, Thuy D. Do, Dominik F. Vollherbst, Philippe L. Pereira, Götz M. Richter, Michael Faerber, Karl H. Weiss, Arianeb Mehrabi, Hans U. Kauczor, Christof M. Sommer

**Affiliations:** 1Clinic for Diagnostic and Interventional Radiology, University Hospital Heidelberg, 69120 Heidelberg, Germany; uh_fengpan@hust.edu.cn (F.P.); thuy.do@med.uni-heidelberg.de (T.D.D.); dominik.vollherbst@med.uni-heidelberg.de (D.F.V.); michael-faerber@gmx.de (M.F.); hu.kauczor@med.uni-heidelberg.de (H.U.K.); 2Department of Radiology, Union Hospital, Tongji Medical College, Huazhong University of Science and Technology, Wuhan 430022, China; 3Department of Neuroradiology, University Hospital Heidelberg, 69120 Heidelberg, Germany; 4Clinic for Radiology, Minimally-Invasive Therapies and Nuclear Medicine, SLK Kliniken Heilbronn GmbH, 74078 Heilbronn, Germany; Philippe.Pereira@slk-kliniken.de; 5Clinic for Diagnostic and Interventional Radiology, Stuttgart Clinics, Katharinenhospital, 70174 Stuttgart, Germany; g.richter@klinikum-stuttgart.de; 6Department of Gastroenterology, University of Heidelberg, 69117 Heidelberg, Germany; KarlHeinz.Weiss@med.uni-heidelberg.de; 7Department of General, Visceral and Transplantation Surgery, University of Heidelberg, 69117 Heidelberg, Germany; arianeb.mehrabi@med.uni-heidelberg.de

**Keywords:** ethiodized oil, electroporation, ablation techniques, hepatocellular carcinoma, X-ray computed tomography

## Abstract

**Simple Summary:**

Irreversible electroporation (IRE) is an effective alternative for the ablation of small hepatocellular carcinoma (HCC) less than 2 cm, which is often poorly visible under unenhanced computed tomography (CT) and/or an ultrasound resulting in the difficulties of complete ablation. In this study, to achieve successful ablation on these small target HCCs with poor invisibility, the combination of transarterial ethiodized oil tumor marking with sequential computed tomography (CT)-guided IRE was performed. After marking, all 11 target-HCCs demonstrated complete visualization in post-marking CT, which were invisible in pre-marking CT. Technically successful ablation was achieved in all sequential IRE procedures. In the follow-up, no residual unablated tumor was observed and the two-year local tumor progression was 27.3%. Thus, ethiodized oil tumor marking with sequential CT-guided IRE is a safe and feasible combination to treat small HCC which was invisible in unenhanced CT.

**Abstract:**

Introduction. To explore the feasibility, safety, and efficiency of ethiodized oil tumor marking combined with irreversible electroporation (IRE) for small hepatocellular carcinomas (HCCs) that were invisible on unenhanced computed tomography (CT). Methods. A retrospective analysis of the institutional database was performed from January 2018 to September 2018. Patients undergoing ethiodized oil tumor marking to improve target-HCC visualization in subsequent CT-guided IRE were retrieved. Target-HCC visualization after marking was assessed, and the signal-to-noise ratios (SNRs) and contrast-to-noise ratios (CNR) were compared between pre-marking and post-marking CT images using the paired *t*-test. Standard IRE reports, adverse events, therapeutic endpoints, and survival were summarized and assessed. Results. Nine patients with 11 target-HCCs (11.1–18.8 mm) were included. After marking, all target-HCCs demonstrated complete visualization in post-marking CT, which were invisible in pre-marking CT. Quantitatively, the SNR of the target-HCCs significantly increased after marking (11.07 ± 4.23 vs. 3.36 ± 1.79, *p* = 0.006), as did the CNR (4.32 ± 3.31 vs. 0.43 ± 0.28, *p* = 0.023). In sequential IRE procedures, the average current was 30.1 ± 5.3 A, and both the delta ampere and percentage were positive with the mean values of 5.8 ± 2.1 A and 23.8 ± 6.3%, respectively. All procedures were technically successful without any adverse events. In the follow-up, no residual unablated tumor (endpoint-1) was observed. The half-year, one-year, and two-year local tumor progression (endpoint-2) rate was 0%, 9.1%, and 27.3%. The two-year overall survival rate was 100%. Conclusions. Ethiodized oil tumor marking enables to demarcate small HCCs that were invisible on unenhanced CT. It potentially allows a safe and complete ablation in subsequent CT-guided IRE.

## 1. Introduction

Percutaneous thermal ablation, such as radiofrequency ablation (RFA), is recommended for patients with hepatocellular carcinoma (HCC) who are unable to undergo surgical resections [1]. Especially for small HCCs with a diameter of less than 2 cm, it is a first-line therapy that could reach a comparable survival rate with resection [2,3]. Irreversible electroporation (IRE), as a non-thermal local tumor therapy can be another effective alternative for the treatment of small HCC during the early stage. However, the clinical evidence is still limited [4,5]. Being different from thermal ablation techniques, IRE was hardly affected by the “heat-sink” effect and could prevent damage of the adjacent thermosensitive structures, such as bile ducts, gallbladder, and a hepatic capsule [5,6,7,8]. The extracellular matrix is rarely affected by IRE, which maintains the potential for regeneration and healing of the liver [7,8]. Thus, IRE can become an alternative option when thermal ablation is contra-indicated [9].

No matter which guidance modality was chosen in percutaneous HCC ablation, adequate visualization of the target tumor under guidance is essential [10,11]. However, small target HCCs less than 2 cm are often poorly visible under unenhanced computed tomography (CT) and/or an ultrasound, especially when the lesions are close to the diaphragm or under the cirrhosis milieu [12,13,14,15,16]. To visualize these small HCCs in an ablation procedure, transarterial ethiodized oil tumor marking was a useful option that does not damage the surrounding healthy liver [12,13,17]. In clinical practice, this ethiodized oil-based visualization could facilitate subsequent percutaneous ablations under CT guidance and, thus, improve the oncological prognosis [15,18,19]. However, it still lacks evidence supporting the benefit of ethiodized oil-based visualization with the following CT-guided IRE in treating small HCCs.

One previous study explored the effect of several different non-metallic embolized agents on IRE in vitro and the result showed an 8.7% reduction of the IRE zone after injecting Embozene^TM^ 500 into the center of the IRE zone [20]. It indicated some types of embolized agents might hamper the IRE by affecting the electric field distribution. Since ethiodized oil is an electric insulator able to increase soft-tissue resistance, it likely reduces the efficacy of subsequent IRE after transarterial injection, resulting in an incomplete ablation. Up to now, there has been no clinical study revealing the effect of embolized ethiodized oil on IRE in HCC patients. Thus, in the present study, the aim was to explore the feasibility and efficiency of an ethiodized oil tumor marking combined with CT-guided IRE for small HCC with poor conspicuity at conventional unenhanced CT.

## 2. Materials and Methods

According to the guidelines of the local ethics committee and/or national research committee along with the 1964 Helsinki Declaration and its later amendments or comparable ethical standards, approval was given by our institute for this study (Ethikkomission der medizinischen Fakultät Heidelberg, Reference Number: S-442/2019) with an analysis of anonymous data, and its completion was also pursuant of best clinical practice.

### 2.1. Treatment Strategy and Inclusion Criteria

In this center, percutaneous locoregional ablation for HCCs was suggested after reaching consensus in the interdisciplinary team meeting regarding the oncologic treatment strategy if there are existing contraindications of surgical resection. Percutaneous thermal ablations under CT/ultrasound guidance including RFA and microwave ablation (MWA) was the first-line treatment following guidelines [1]. IRE was chosen instead of other thermal ablations only when target HCC were closed to (<1.0 cm) vasculature and/or other critical structures including bile ducts, gallbladder, diaphragm, heart, and kidney, or after previous partial hepatectomy [1,9]. Prior transarterial chemo-embolization (TACE) was recommended to perform in this center owing to the association with significantly higher overall and recurrence-free survival than single ablation [1]. If the target HCCs were smaller than 2.0 cm with liver cirrhosis manifestation or after a previous partial hepatectomy, a pure ethiodized oil tumor marking was performed to avoid liver damage before ablation.

In this study, consecutive patients between January 2018 and September 2018 were retrieved from the institutional digital databases and the GE PACS and Centricity RIS databases (GE Medical Systems, Buckinghamshire, UK). The inclusion criteria were defined as follows: 1. patients with primary or de novo HCC (hereinafter target-HCC) according to the Liver Imaging Reporting and Data System 2018 (LI-RADS 2018) diagnostic criteria [21]; 2. pure ethiodized oil tumor marking following IRE was performed; 3. no previous regional or systematic therapy for the target-HCC (e.g., TACE, Sorafenib, etc.); 4. this novel combination therapy was accepted by patients based on intention-to-treatment principle and informed consents were acquired before procedures.

### 2.2. Pre-Interventional Imaging

Standardized risk stratification was undertaken based on the contrast-enhanced magnetic resonance imaging (MRI) (MAGNETOM Aera 1.5 T, Siemens, Erlangen, Germany), according to the LI-RADS 2018 at the University Hospital Heidelberg, Germany, owing to the comprehensive imaging algorithm with high specificity for HCC [21,22,23]. For a description of the index tumor size, the maximum HCC diameter was measured on an axial MRI at the phase where the margins were most clearly demarcated by one radiologist (with more than eight years of experience in abdominal radiology) [23]. Unenhanced pre-marking CT (SOMATOM Definition AS; Siemens Healthineers, Erlangen, Germany) was performed with the parameter setup as: single energy technique, and tube voltage of 120 kVp with an adaptive current-time product. The CT images were transversely reconstructed with a slice thickness of 1.5 mm and an overlap of 1.5 mm using a soft-tissue kernel (B30 f; SiemensMedical Solutions, Siemens, Forchheim, Germany). These pre-interventional radiological examinations were performed before marking within 1 month (Figure 1A,B).

### 2.3. Ethiodized Oil Tumor Marking Procedure

The ethiodized oil tumor marking was performed using an Artis Zee angiography system (Siemens Healthineers, Erlangen, Germany). After the peri-interventional medication (250 mg prednisolone, 4 mg ondansetron, 1.25 mg midazolam, 1 g novamine, and 7.5 mg piritramide), a 5 F sheath (Terumo, Tokyo, Japan) was inserted via the left or right common femoral artery by using the Seldinger technique under local anesthesia. A 5 F sidewinder catheter (Cordis, Miami, FL, USA) was placed in the celiac trunk after configuration in the aortic arch (Figure 1C). After digital subtraction angiography (DSA) for overview imaging, a coaxial 2.0, 2.4, or 2.8 F microcatheter (Progreat, Terumo, Tokyo, Japan; or Direxion, Boston Scientific, Malborough, MA, USA) was navigated into the common hepatic artery. Afterward, a trans-microcatheter contrast-enhanced cone-beam CT was performed to locate the target-HCC including its arterial branches [24] (Appendix A and Figure 1D). Then, the microcatheter was advanced into the definite subsegmental arterial branch supplying the target-HCC and a DSA was performed. After confirming the target-HCC by means of a hyper-vascularized focal tumor blush, ethiodized oil (Lipiodol, Guerbet, Villepinte, France) was injected slowly with a velocity of 1–3 mL/min ensuring antegrade propagation. The injection endpoint was defined as ethiodized oil accumulation within the target-HCC and peripheral stasis in the supplying arterial branch as documented 5 min after ethiodized oil injection. After the injection of ethiodized oil, the cone-beam CT was performed with the same setting to check the opacity of the target-HCC (Figure 1E).

### 2.4. IRE Treatment

One day after marking, IRE was performed under single CT-guidance (SOMATOM Definition AS, Siemens Healthineers, Erlangen, Germany) because of invisibility of the small target HCCs under ultrasound (Acuson S2000 US system, Siemens Healthcare, Erlangen, Germany) by using a 4 V1 vector array probe (1–4.5 MHz). A commercial IRE system (NanoKnife Electroporator, AngioDynamics, Queensbury, NY, USA) was used following the standard practice recommendation [25]. Under general anesthesia with complete muscle relaxation and artificial ventilation, an unenhanced CT scan (post-marking CT) of the liver was carried out after the placement of a radiopaque optical marker on the patient’s skin to define the target-HCC (Figure 1F) with the same CT parameters as the pre-marking CT. Then, the target-HCCs were identified with the assistance of both the ethiodized oil opacity and the previous enhanced MRI. Conventional 17 G monopolar electrodes (NanoKnife; AngioDynamics, Queensbury, NY, USA) were used in all the IRE procedures. The recommended IRE electrodes configuration and ablation settings for different tumor sizes were calculated using the Nanoknife planning software (Procedure Manager V2.2.0.23, AngioDynamics, Queensbury, NY, USA) [10]. After manual electrode insertion, a needle-position control CT scan was performed to confirm the placement of the electrodes (Figure 1G). Afterward, IRE was performed with electrocardiographically (ECG) synchronized pulse delivery according to the manufacturer’s instructions. After completion of the pulse applications, contrast-enhanced CT was performed to confirm if the IRE zone covered the entire target-HCC with a sufficient ablative margin (≥5 mm) (Figure 1H).

### 2.5. Radiological Follow-Up

Contrast-enhanced liver MRI was performed under the same machine as mentioned above. After IRE, regular MRI was carried out with three-month intervals lasting for two years.

### 2.6. Study Endpoints

#### 2.6.1. Success of Ethiodized Oil Tumor Marking

Based on the distribution of the ethiodized oil in the post-marking CT, three grades of visualization were defined: 1. Complete visualization—complete opacification of the target-HCC by ethiodized oil accumulation with a definite delineation of the tumor margin (Figure 1); 2. Incomplete visualization—a clear defect of ethiodized oil accumulation in the target-HCC but still being able to localize the tumor; 3. No visualization—no clear contrast between the target-HCC and the surrounding normal liver tissue [26,27]. Technical success was defined as complete or incomplete visualization. All CT images were interpreted by two radiologists together (C.M.S. and F.P. with more than ten and eight years of experience in abdominal radiology, respectively).

#### 2.6.2. Quantitative Analysis of the Visualization of Target-HCCs

The Hounsfield scale of the target-HCCs, the surrounding normal liver tissue (the region adjacent to the target-HCC with visually highest density), the peripheral normal liver tissue, and paravertebral muscles were manually measured by using circular or oval regions-of-interest (ROIs) on the pre-marking and post-marking CT images. Large vessels and prominent artifacts were carefully avoided. An exemplary measurement is shown in Figure 2. The signal-to-noise ratios (SNRs) and contrast-to-noise ratios (CNR) were evaluated by using the formulas below.

SNR = CT_target_/SD_noise_, in which CT_target_ referred to the mean Hounsfield scales of the estimated tissue, and the SD_noise_ was the standard deviation for Hounsfield scales of the paraspinal muscles [28];CNR = ∣(CT_HCC_ − CT_liver_)∣/SD_noise_, in which CT_HCC_ referred to the Hounsfield scale of the target-HCC, CT_liver_, and SD_noise_ referred to the mean and standard deviation of Hounsfield scale of the surrounding normal liver tissue [29].

The CT measurements were independently measured by two radiologists, as mentioned above. Each measurement was performed twice per observer, as shown in Reading 1 and Reading 2.

#### 2.6.3. Success of IRE

The IRE reports were summarized based on the standard report criteria [10,25]. The technical success of IRE was defined as the target-HCC that was successfully treated, according to the protocol requirement and covered completely by the IRE zone with at least a 5 mm ablative margin [10,25]. The adverse events were collected as mentioned above.

#### 2.6.4. Adverse Events and Follow-Up

The adverse events of ethiodized oil tumor marking and IRE were collected and classified under the updated standards of the Society of Interventional Radiology [30]. For the radiological follow-up, two major endpoints for local treated HCCs were defined based on LI-RADS 2018: endpoint-1—residual unablated tumor that was defined in the initial follow-up imaging demonstrated to be residual HCC at the ablative margin (LR-TR), endpoint-2—local tumor progression that was defined after at least one contrast-enhanced radiological follow-up study documenting an absence of viable tumor tissue in/around the target-HCC, new HCC foci appeared at the edge of the IRE zone (LR-TR) in further follow-up [21,25,31]. Both endpoint-1 and endpoint-2 were defined as treatment failures [31]. A radiological interpretation of the endpoints was performed by two radiologists together (with more than ten and eight years of experience, respectively, in abdominal radiology). The overall survival was also assessed.

### 2.7. Statistics

Statistical analyses were performed using IBM SPSS Statistics Software (version 26; IBM, New York, NY, USA). Quantitative data were presented as mean ± standard deviation (minimum-maximum), while the counting data were presented as count (percentage of the total). The comparisons of quantitative data were evaluated by using the paired-samples *t*-test, according to the normal distribution assessed by the Shapiro-Wilk test. The tests were two-sided and a *p*-value of <0.05 was defined as the statistical significance. Intra-observer and inter-observer agreements of the measurements were calculated by applying the Bland-Altman analysis with bias and 95% limits of agreement [32].

## 3. Results

### 3.1. Patient Collective and Target-HCC Characteristics

After retrieving 91 HCC patients who underwent transarterial ethiodized oil embolization with sequential ablation, a total of nine patients (seven males and two females), with 11 target-HCCs were finally included (Figure 3). The average age of the patients was 67 ± 5 years (range: 59–76 years). The Child-Pugh grade was A (5 points) and ECOG grade was 0 in all patients. Cirrhosis existed in all patients ascribing to different liver diseases. Five (4) patients (44.4%) had HCC resection histories prior to de novo HCC recurrence. Seven (7) patients had solitary target-HCC, and two (2) patients had dual-target HCCs. No vascular invasion or extrahepatic metastases were found. In the prior MRI examinations, all the targets-HCCs showed clear arterial phase hyperenhancement (APHE). Ten (10) target-HCCs were classified into the LR-5 (definite HCC) by the LI-RADS 2018 grade system, while only one target-HCC was classified into LR-4 (probable HCC), which was confirmed as moderately differentiated HCC by the following CT-guided biopsy [21]. The average size of the target-HCCs was 14.6 ± 3.0 mm (range: 11.1–18.8 mm). The basic characteristics of the patient collective and target-HCCs are summarized in Table 1.

### 3.2. Success of Ethiodized Oil Tumor Marking

No target-HCC was visualized (0/11, 0.0%) in the pre-marking CT scan. In the ethiodized oil tumor marking procedure, the average ethiodized oil usage per target-HCC was 1.9 ± 1.1 mL (range: 1.0–4.0 mL). The post-marking CT demonstrated ideal ethiodized oil accumulation in all target-HCCs (11/11, 100.0%) referring to prior MRI examinations, in terms of complete visualization. The technical success rate was 100%.

### 3.3. Quantitative Analysis of the Visualization of Target-HCCs

The results of the quantitative analysis of the target-HCC visualization are summarized in Table 2. Before marking, the Hounsfield scale of target-HCCs showed only slight differences from the surrounding normal liver tissue (50.6 ± 11.2 vs. 56.9 ± 9.0 HU) with a CNR of 0.43 ± 0.28. After marking, the Hounsfield scale of target-HCC and surrounding liver parenchyma increased significantly (*p* = 0.001 and 0.005, respectively). The SNR and CNR of the target-HCCs significantly increased after marking compared with before marking (11.07 ± 4.23 vs. 3.36 ± 1.79, *p* = 0.006, and 4.32 ± 3.31 vs. 0.43 ± 0.28, *p* = 0.023, respectively). The Bland-Altman analysis showed good intra-observer and inter-observer agreement regarding the Hounsfield scale measurements (Appendix A).

### 3.4. Success of IRE

The standard reports of IRE are summarized in Table 3. The mean operating time between first electrode insertion and last electrode removal was 46 ± 17 min (range: 28–74 min), while the application time of pulse delivery was 15 ± 8 min (range: 6–32 min). For all IRE treatments, 3 ± 1 (2–4) electrodes were punctured in parallel under CT guidance. Electrode pairs were placed at a mean inter-electrode space of 1.5 ± 0.2 cm (range: 1.0–1.8 cm). The pulse length was set as 90 μs with 90 pulses per burst in all procedures. The mean total pulse number per target-HCC was 360 ± 241 (range: 180–990). The average voltage setting was 1835 ± 273 V/cm (range: 1448–2347 V/cm). The current of all treatment pulses was above 20 A, from 24.6 A to 43.6 A with an average of 30.1 ± 5.3 A. In all IRE bursts, both the delta ampere (ΔA burst) and percentage (ΔA% burst) were positive with the mean values of 5.8 ± 2.1 A and 23.8 ± 6.3%, respectively. After IRE, the enhanced CT images in the portal phase demonstrated a large enough IRE zone completely covering each target HCC with a larger than 0.5 mm ablative margin (Table 3 and Figure 4). The technical success rate was 100%.

### 3.5. Adverse Events and Follow-Up

No adverse event after ethiodized oil tumor marking and IRE was reported. Two (2) days after IRE, all nine patients were discharged. In the radiological follow-up, no residual, unablated tumor (endpoint-1) was observed in all treated target-HCCs. The half-year, one-year, and two-year local tumor progression (endpoint-2) rate was 0%, 9.1%, and 27.3%. Treatment success was achieved in 8 HCCs (72.7%) in a two-year follow-up (Table 3 and Figure 4). The overall survival rate was 100%.

## 4. Discussion

In this study, ethiodized oil tumor marking in combination with IRE was performed to treat small HCCs with a size from 11.1–18.8 mm, which were difficult to be clearly defined under unenhanced CT scans [21]. All ethiodized oil tumor markings and IRE procedures were technically successful without any adverse event. After marking, all the target-HCCs could be completely demarcated in post-marking CT during the IRE procedures. In the radiological follow-up, no residual unablated tumor was observed. Three treated HCCs occurred with local tumor progression after a follow-up period of two years. The treatment success rate reached 72.7% and the overall survival rate was 100% after a two-year follow-up.

Although an early definitive diagnosis of small HCC before it exceeds 2 cm in diameter increases the treatment success, these small HCCs are mostly invisible on unenhanced CT, thus, impeding accurate ablation [1,12,13]. Ultrasound guidance as a standard guiding method in percutaneous ablation demonstrates many advantages, such as real-time monitoring and the absence of X-ray radiation, but the invisibility of small HCCs has the same issue [11,33]. From previous studies, the overall detection rate of small HCCs by conventional ultrasound was from 15% to 78% [16]. In this cohort, sole CT guidance was used because of invisibility of small target HCCs under ultrasound after marking. Although contrast-enhanced ultrasound or an ultrasound-based fusion imaging system can increase the detectability of smalls HCCs, it has not been carried out as widely as CT guidance and is still difficult to accurately localize some small subcapsular or subphrenic HCCs, especially when pseudo-lesions were surrounded [3,34]. To improve the visualization of small HCCs, the combination of ethiodized oil tumor marking with sequential CT-guided percutaneous ablation appears to be another feasible option [15,26]. In this study, ethiodized oil tumor marking provided an accurate visualization of the target lesions and facilitate precise electrode configuration afterward. Besides, the ablation zone looked like a de-vascularized “egg” with a “hyperdense yolk” in the center (target HCC after marking) under enhanced CT immediately after IRE in this cohort. Therefore, it was much easier for the operator to identify if there was enough of a safety margin. Although TACE with sequential IRE might bring a better prognosis owing to the increase of intracellular chemotherapy concentration, in order to objectively estimate the effects of ethiodized oil on sequential IRE, only patients undergoing marking with pure ethiodized oil injection were included in this study because of its lack of an anti-tumor effect [1,35,36].

Being different from thermal ablation techniques, multiple electrode configurations are recommended for IRE in the HCC treatment [4,10]. The precise arrangement of these electrodes, according to the tumor geometry, is very important because a very small error of the electrode configuration will bring a very irregular electric field distribution leading to heterogenous and incomplete ablation [37,38]. That’s why, in IRE practice, CT guidance was more recommended than ultrasound guidance because of the direct 3D visualization of the target HCC and electrode array [10,39]. In this study, all the small target-HCCs were not visible in pre-marking CT scans, which were accompanied by a very low SNR and CNR (3.36 ± 1.79 and 0.43 ± 0.28, respectively). However, after marking, the unenhanced CT showed complete opacities of the target-HCCs with definite delineations of the tumor margin, forming a clear contrast to the surrounding normal liver tissue. Quantitatively, both the SNR and CNR of the target-HCC (11.07 ± 4.23 and 4.32 ± 3.31, respectively) significantly increased after marking. Afterward, this complete visualization provided an objective morphology of the target-HCC for the CT-guided IRE, ensuring the precise configuration of the multiple electrodes.

In a previous in vivo experiment in pig livers, it was found that ethiodized oil embolization of the healthy liver does not bring any adverse effect to the sequential IRE, including the electric property changes, IRE zone geometry, and histopathological features [40]. This study following the previous work showed no adverse effect of ethiodized oil marking on subsequent IRE in HCC patients [40,41]. Both the ΔA burst and ΔA% burst were positive in all IRE bursts (average: 5.8 ± 2.1 A and 23.8 ± 6.3%, respectively) and the current was above 20 A with an average of 30.1 ± 5.3 A. These records indicated ideal IRE effects in the procedures regarding all target-HCCs after marking [10,42]. As a result, the post-IRE CT demonstrated a complete covering of the IRE zone in each target-HCC with a sufficient ablative margin.

In addition, no adverse event was reported in all ethiodized oil tumor marking and IRE procedures. All the patients tolerated the procedures well. Furthermore, no needle track seeding was found in the radiological follow-up. In one study, the incidence of needle tract seeding after IRE was reported to be 26% [43]. It is speculated that the complete visualization of the target-HCCs by ethiodized oil tumor marking can reduce the needle track seeding complications by facilitating the electrode puncture because the multiple puncture attempts and the frequent pull-back maneuver are the major reasons that cause the needle tract seeding [43,44].

In previous clinical studies, incomplete ablation rates of HCC after IRE concentrated around 22% with local progression rates of 5–30% during a follow-up period of 1–2 years [42,43,45]. However, in the present study, no residual unablated tumor was detected in the radiological follow-up. It might be also ascribed to ethiodized oil tumor marking facilitating a more precise configuration of the IRE electrodes, which resulted in a higher complete ablation rate. However, three local tumor progressions (27.3%) occurred in a two-year follow-up without the incidence of a death event, which was similar to the previous literature [42,43,45]. It was speculated that local electrical heterogeneity of the tumor tissue resulted in small “live tumor patches” after IRE, leading to a mid-term or long-term local recurrence [4,46].

In summary, this study suggests that the ethiodized oil-based tumor marking with sequential IRE is a feasible therapeutic strategy for the treatment of small HCCs invisible on unenhanced CT. The complete visualization after marking could reduce the difficulties regarding the electrode puncture and configuration, and would avoid incomplete ablations and needle tract seeding complications. However, due to the inhomogeneous IRE in tumor tissue, this combination treatment might not be enough to reduce the local tumor progression rate. As a speculation, using therapeutical TACE instead of ethiodized oil tumor marking with sequential IRE likely brings better long-term local control in the treatment of small HCCs, just like the merits from the combination of TACE with RFA [1].

This study has several limitations. The first limitation is the small group of patients with limited follow-up periods because IRE was a second-line ablation for small HCC with very strict indications in this center. Second, there were several potential biases and confounders in this study, such as both patients with an initial diagnosed HCC and prior HCC resection history were included, which might bring some bias in estimating the prognosis. Thus, only the primary and de novo HCCs without either the previous local or systematic treatment were involved, and the endpoints about the local tumor control were assessed. LI-RADS estimation was used instead of Modified Response Evaluation Criteria in Solid Tumors (mRECIST) owing to its advantage on a local tumor assessment. Third, there was no control group of single CT-guided IRE treatment without prior ethiodized oil tumor marking because the small target HCCs were invisible under unenhanced CT, resulting in the impossibility of single IRE performance.

## 5. Conclusions

It is sometimes difficult to clearly define the small HCCs on unenhanced CT. Ethiodized oil tumor marking enables complete visualization of these small HCCs in subsequent CT-guided IRE but without an adverse impact on the IRE effect. This combination treatment can facilitate the performance of the subsequent CT-guided IRE and potentially avoid incomplete ablations and needle tract seeding complications.

## Figures and Tables

**Figure 1 cancers-13-02021-f001:**
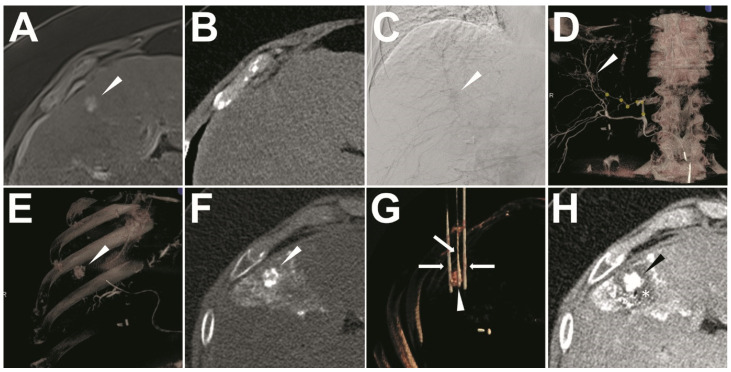
The exemplary illustration of the ethiodized oil tumor marking with sequential IRE treatment. A target-HCC (arrowhead) close to the diaphragm with typical arterial APHE was confirmed in enhanced MRI (**A**), which was invisible in pre-marking unenhanced CT (**B**). In the ethiodized oil tumor marking procedure, the opacity of the target-HCC was found in the celiac artery angiography (**C**). Then, the angiographic cone-beam CT was performed to navigate the micro-catheterization into the definite feeding artery (yellow round marks) of the target-HCC (arrowhead) in the 3D rendering image (**D**). After successful subsegmental micro-catheterization and ethiodized oil injection, the cone-beam CT was reperformed demonstrating visualization of target-HCC (arrowhead) (**E**). In post-marking CT, the target-HCC (arrowhead) could be defined visually with complete ethiodized oil accumulation (complete visualization) (**F**). In the subsequent IRE procedure, three (3) electrodes (arrow) were inserted surrounding the target-HCC (arrowhead) with a triangle configuration (**G**). Immediately after IRE, the enhanced CT scan was performed to identify the de-vascularized IRE zone (*) in the portal vein phase (**H**).

**Figure 2 cancers-13-02021-f002:**
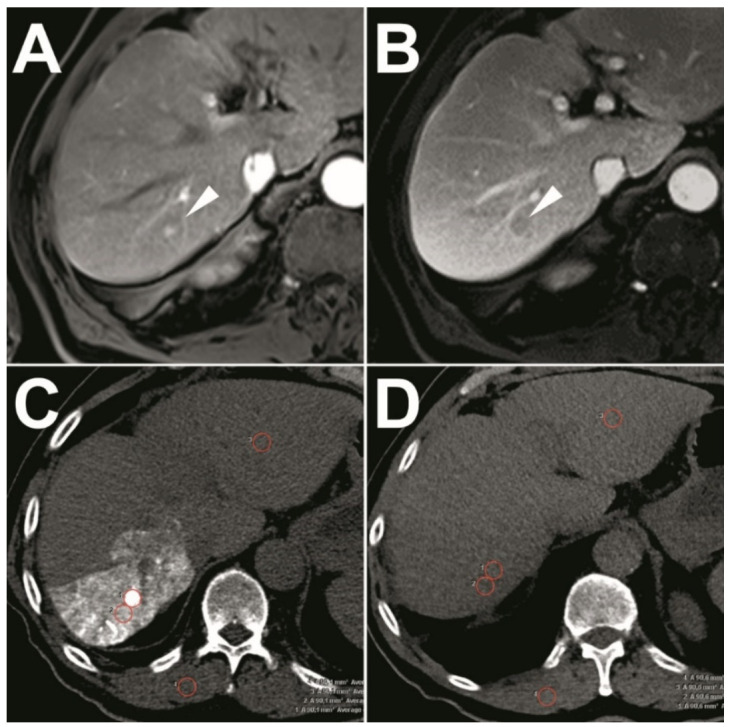
An exemplary illustration of the quantitative Hounsfield scale measurements. A target-HCC with typical arterial APHE (**A**) and “washout” in the portal phase (**B**). Branches of portal veins surrounding the lesions were observed (**B**) so the IRE was chosen in avoidance of “heatsink” if thermal ablation was performed. In the post-marking CT images, the target-HCC (arrowhead) was opacified, which was defined as complete visualization (**C**). In the Hounsfield scale measurements, the circular or oval region-of-interests (ROIs) (red circles) manually covered the entire target-HCCs (**1**), the surrounding normal liver tissue (**2**-the region adjacent to the target-HCC with visually highest density), the distant normal liver parenchyma (**3**), and paravertebral muscles (**4**) with the same size and shape. Large vessels and prominent artifacts were avoided carefully. The identical measurements with the same rules were performed on the pre-marking CT images (**D**). However, because the target-HCC could not be defined clearly in the pre-marking CT images (**D**), the pre-interventional MRI images and post-marking CT images could be used as a reference for the measurements.

**Figure 3 cancers-13-02021-f003:**
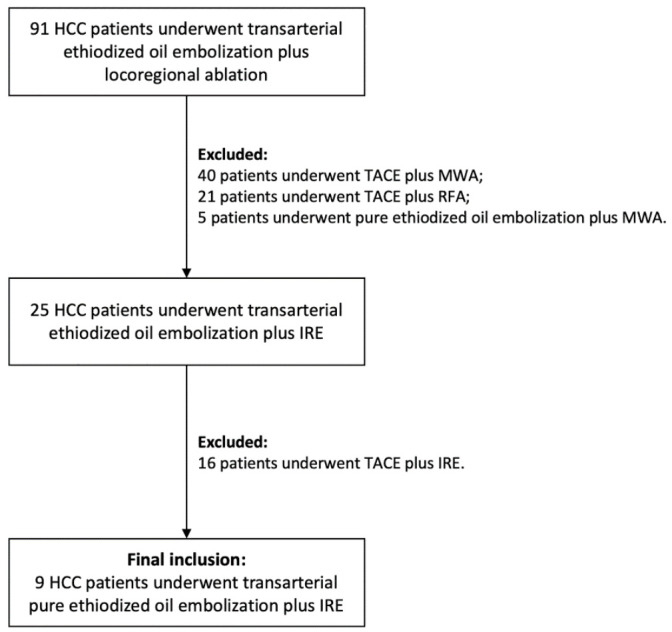
Flowchart of patients inclusion.

**Figure 4 cancers-13-02021-f004:**
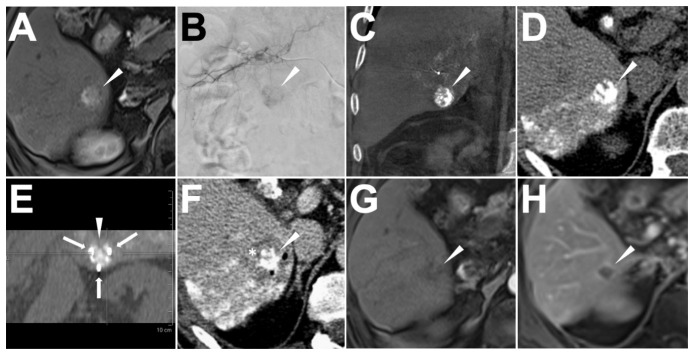
A patient with a de novo HCC close to the right kidney in segment VI with a size of 17.8 mm. A subcapsular target-HCC (arrowhead, LR-5, No. 10) with an increasing size of 17.8 mm at Seg VI of the liver was confirmed with definite APHE in the enhanced MRI scan (**A**). In the ethiodized oil tumor marking procedure, a clear opacity (arrowhead) of the target-HCC was found in the subsegmental angiography (**B**). After slowly injecting 1.3 mL Lipiodol for the marking, the cone-beam CT showed a completely local Lipiodol accumulation in the target-HCC (arrowhead) (coronal axis, (**C**)). One day later, IRE was performed. In the post-marking CT, the target-HCC (arrowhead) could be clearly defined as complete visualization (**D**). With three electrodes (arrow) putting in a triangle configuration around the target-HCC (arrowhead) with tip exposures of 2.0 cm and a mean electrode pair distance of 1.5 cm (**E**), three (3) IRE bursts with a total of 270 pulses were applied with an average current of 43.6 A. The post-IRE enhanced CT scan showed a de-vascularized rim (*) as the ablative margin surrounding the target-HCC (arrowhead) in the portal vein phase (**F**). After a follow-up of two years, the enhanced MRI follow-up showed a persistent de-vascularized IRE zone (arrowhead) in the arterial phase (**G**) and portal phase (**H**) with shrinkage involution.

**Table 1 cancers-13-02021-t001:** Patient collective and target-HCC characteristics.

Patients No.	Etiology of Cirrhosis	Child-Pugh Grade	ECOG Grade	Prior HCC History	Prior HCC Resection History	Target-HCC No.	Location of Target-HCC (Segment)	HCC Size (mm)	APHE	Enhancing “Capsule”	Non-Peripheral “Washout” in Portal Phase	Threshold Growth *	LI-RADS Grades	Pathological Results of the Target Tumor
1	NASH	A	0	Yes	Yes	1	7	15.5	Yes	No	No	Yes	LR-5	
2	Chronic hepatitis C	A	0	No	No	2	8	13.5	Yes	No	Yes	No	LR-5	
						3	8	11.6	Yes	No	Yes	Yes	LR-5	
3	Ethytotoxic cirrhosis	A	0	Yes	Yes	4	5	11.2	Yes	No	Yes	Yes	LR-5	
4	Ethytotoxic cirrhosis	A	0	No	No	5	4a	15.4	Yes	No	Yes	Yes	LR-5	
5	PSC	A	0	Yes	Yes	6	5	15.7	Yes	No	Yes	Yes	LR-5	
						7	8	18.7	Yes	No	Yes	Yes	LR-5	
6	Ethytotoxic cirrhosis	A	0	No	No	8	4a	18.8	Yes	No	No	No	LR-4	Moderately- differentiated HCC
7	Chronic hepatitis C and B	A	0	Yes	Yes	9	6	11.5	Yes	No	Yes	Yes	LR-5	
8	Chronic hepatitis B	A	0	No	No	10	6	17.8	Yes	No	No	Yes	LR-5	
9	Chronic hepatitis C	A	0	No	No	11	8	11.1	Yes	No	Yes	No prior examination	LR-5	

NASH—nonalcoholic steatohepatitis. PSC—primary sclerosing cholangitis. APHE—arterial phase hyperenhancement. * Threshold growth was defined as a mass size increase of ≥50% in ≤6 months [21].

**Table 2 cancers-13-02021-t002:** The quantitative analysis of the visualization of target-HCCs.

CT Measurement Parameters	Pre-Marking CT	Post-Marking CT	*p*-Value *
Area of ROIs (mm^2^)	147.4 ± 90.0 (60.0–332.7)	148.6 ± 97.2 (49.1–331.8)	0.445
Hounsfield scale of the target-HCCs (HU)	50.6 ± 11.2 (34.8–65.9)	206.4 ± 68.0 (113.4–350.5)	**0.001**
Hounsfield scale of the surrounding normal liver tissue (HU)	56.9 ± 9.0 (40.5–67.1)	94.4 ± 22.6 (68.4–133.0)	**0.005**
Hounsfield scale of the peripheral normal liver tissue (HU)	59.4 ± 8.5 (43.7–66.6)	58.2 ± 9.1 (45.2–69.7)	0.130
SNR of the target-HCCs	3.36 ± 1.79 (1.92–7.48)	11.07 ± 4.23 (4.02–19.52)	**0.006**
SNR of the surrounding normal liver tissue	3.74 ± 1.72 (2.66–7.63)	5.09 ± 1.81 (2.79–8.29)	0.087
SNR of the peripheral normal liver tissue	3.74 ± 1.59 (2.58–7.10)	3.13 ± 0.98 (1.87–5.04)	0.258
CNR of the target-HCCs	0.43 ± 0.28 (0.11–0.88)	4.32 ± 3.31 (1.44–13.01)	**0.023**

* Paired *t*-test. The bold number of *p*-value indicated the statistical significance.

**Table 3 cancers-13-02021-t003:** Reports of IRE treatment and follow-up.

Target-HCC No.	Operating Time between First Electrode Insertion and Last Electrode Removal (min)	Application Time of the IRE Pulse Delivery (min)	Index Tumor Size (Depth/ Width/ Length) (mm)	Number of Electrodes	Electrode Configuration	Active Tip Length (cm)	Average Inter-Electrode Space (cm)	Number of Electrode Pairs	Pulse Length (μs)	Number of IRE Burst	Average Pulse Number per IRE Burst	Total Pulse Number	Average Voltage Setting (V/cm)	Average Current (A)	Energy (J)	Average ΔA Burst (Minimum-Maximum) (A)	Average ΔA% burst (Minimum-Maximum) (%)	IRE Zone (Depth/Width/Length) (mm)	Technical Success of IRE	Treatment Failure (Endpoint-1 and -2) *	Period until Endpoint-2 * (Months)
1	62	9	16.5/15.7/ 15.1	3	Triangle	2	1.5	3	90	3	90	270	1589	25.4	358.2	4.5 (3.1–5.3)	20.0 (15.3–23)	32.7/30.3/31.0	Success		
2	34	14	13.7/14.5/ 13.9	3	Triangle	2	1.4	3	90	4	90	360	2000	30.1	506	5.3 (3.2–6.4)	21.1 (11.4–25.5)	31.6/29.7/30.1	Success		
3	57	10	12.9/12.3/ 11.5	3	Triangle	2	1.4	3	90	3	90	270	1817	29.9	457.1	4.2 (2.3–5.4)	17.6 (10.5–22.6)	32.8/30.1/31.2	Success		
4	33	17	12.5/11.9/ 11.7	3	Triangle	2	1	3	90	3	90	270	2347	26.6	380.8	5.4 (3.1–6.6)	25.0 (15.2–33.8)	32.1/25.2/26.9	Success		
5	29	11	16.7/15.7/ 16.1	3	Triangle	2	1.5	3	90	3	90	270	1448	24.6	322.7	2.9 (1.9–3.6)	14.2 (9.2–17.5)	33.5/30.5/31.5	Success	Local tumor progression	11 ^#^
6	30	10	16.5/17.1/ 16.2	3	Triangle	2	1.6	3	90	3	90	270	1875	31.1	552.2	6.7 (5.9–7.8)	27.9 (24.5–32.4)	31.5/32.7/30.6	Success		
7	74	32	19.9/18.7/ 18.2	4	Square	2.5	1.8	6	90	11	90	990	1580	30.1	522.6	7.0 (2.8–11)	30.4 (11.8–52.9)	37.6/35.5/35.8	Success	Local tumor progression	14 ^#^
8	61	28	19.2/19.7/ 18.8	4	Square	2.5	1.8	6	90	7	90	630	1525	34.7	547.4	5.5 (1.6–10.1)	22.2 (5.5–50.8)	36.2/37.8/35.3	Success		
9	28	8	10.1/0.91/ 0.87	2	Line	2	1.5	1	90	2	90	180	2000	26.7	481.3	4.4 (4.2–4.5)	18.7 (17–20.5)	29.5/22.9/23.9	Success		
10	57	15	18.8/18.1/ 18.5	3	Triangle	2	1.5	3	90	3	90	270	2006	43.6	756.5	10.6 (9.1–13.5)	32.2 (27.6–40.9)	33.4/34.5/32.7	Success		
11	37	6	10.3/0.95/ 0.91	2	Line	2	1.5	1	90	2	90	180	2000	28.7	406.4	7.7 (3.8–11.5)	33.0 (17–48.9)	29.9/23.1/23.5	Success	Local tumor progression	17 ^#^
Average	46 ± 17	15 ± 8		3 ± 1		2.0 ± 0.2	1.5 ± 0.2	3 ± 2	90 ± 0	4 ± 3	90 ± 0	360 ± 241	1835 ± 273	30.1 ± 5.3	481.0 ± 120.0	5.8 ± 2.1	23.8 ± 6.3				
	(28–74)	(6–32)		(2–4)		(2.0–2.5)	(1.0–1.8)	(1–6)	(90–90)	(2–13)	(90–90)	(180–990)	(1448–2347)	(24.6–43.6)	(322.7–756.5)	(2.9–10.6)	(14.2–33.0)				

* Based on LI-RADS 2018, two major endpoints were defined: endpoint-1—residual unablated tumor was defined as the initial follow-up imaging demonstrated residual HCC at the ablative margin (LR-TR). Endpoint-2—local tumor progression was defined as after at least one contrast-enhanced follow-up study documenting an absence of viable tumor tissue in/around the target-HCC, new HCC foci appeared at the edge of the IRE zone (LR-TR) [21,25,31]. Both endpoint-1 and endpoint-2 were defined as treatment failures [31]. ^#^ The period between the performance of IRE and the time of endpoint-2 being identified.

## Data Availability

The data presented in this study are available on reasonable request from the corresponding author. The data are not publicly available due to the protection of the participants’ privacy and ethical restriction.

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
