# Peer review of "Percutaneous Irreversible Electroporation for Treatment of Small Hepatocellular Carcinoma Invisible on Unenhanced CT: A Novel Combined Strategy with Prior Transarterial Tumor Marking"

_cancers, 2021, doi:10.3390/cancers13092021_

Round 1
Reviewer 1 Report
Any
Reviewer 2 Report
The revised article is very interesting and can be useful in the clinical field.
This manuscript is a resubmission of an earlier submission. The following is a list of the peer review reports and author responses from that submission.
Round 1
Reviewer 1 Report
none
Reviewer 2 Report
I read the reply from the authors . Although some raised questions have been addressed in their covering letter, some unanswered relevant issues still remain. In particular, although the authors included a flow chart figure, it is still hard understanding why they decided to perform ethiodized oil tumor marking in 91 patients most of whom underwent RF or MWA. The authors should explain this point and specify what do they mean by " intention-to treat by patient". In addition, in the discussion, authors still stated that Ultrasound guidance in an "alternative....". In my opinion this sentence should be changed as " ultrasound guidance is the standard of care in percutaneous ablation".
I would like to reiterate that, in my personal opinion, the manuscript looks more like a technical report rather than a clinical report and I'm still convinced that the "Cancers" is nor the most proper Journal where submitting it.
Reviewer 3 Report
The authors, Feng Pan et al. suggested Ethiodized oil tumor marking enables to demarcate small HCCs that were invisible on unenhanced CT.
The authors described its treatment techniques, so far, thermal
ablation was still firstly recommended, as well as in their center.
RFA is usually performed using abdominal echo and if the nodules are difficult to detect contrast-enhanced echo(sonazoid, etc) may be used to identify the lesion.
Since there is no description of these techniques, again in revised manuscript, there are no novel enough to be useful technology.